# Four Markers Useful for the Distinction of Intrauterine Growth Restriction in Sheep

**DOI:** 10.3390/ani13213305

**Published:** 2023-10-24

**Authors:** Wannian Wang, Sijia Chen, Liying Qiao, Siying Zhang, Qiaoxia Liu, Kaijie Yang, Yangyang Pan, Jianhua Liu, Wenzhong Liu

**Affiliations:** 1Department of Animal Genetics, Breeding and Reproduction, College of Animal Science, Shanxi Agricultural University, Taigu, Jinzhong 030801, China; wannian1876@163.com (W.W.); chensijia8297@163.com (S.C.); liyingqiao1970@163.com (L.Q.); zhangsy9526@163.com (S.Z.); kjyang@sxau.edu.cn (K.Y.); panyy@sxau.edu.cn (Y.P.); ljhbeth@163.com (J.L.); 2Shanxi Animal Husbandry Technology Extension Service Center, Taiyuan 030001, China; zzzlqx@126.com; 3Key Laboratory of Farm Animal Genetic Resources Exploration and Breeding of Shanxi Province, Jinzhong 030801, China

**Keywords:** IUGR, sheep, related diagnostic genes, potential markers, machine learning algorithms

## Abstract

**Simple Summary:**

Intrauterine growth restriction (IUGR) is a disruption in the development of animal embryos or fetal organs in the maternal uterus and is highly regarded because of its serious detrimental effects on animal husbandry. However, there are few diagnostic methods and treatment options for IUGR, and its molecular mechanisms are not fully understood. In this study, a bioinformatics approach was applied to identify four IUGR-related diagnostic genes (IUGR-DGs) in sheep, construct a gene scoring system based on the IUGR-DGs, and evaluate the relationship between the IUGR score and disease risk. A new artificial neural network model was constructed to better diagnose the occurrence of IUGR in sheep based on the four IUGR-DGs. Although future studies will be necessary to elucidate the entire molecular mechanisms involved, we believe that the findings of this paper will facilitate the diagnosis and treatment of IUGR in sheep and improve livestock production levels.

**Abstract:**

Intrauterine growth restriction (IUGR) is a common perinatal complication in animal reproduction, with long-lasting negative effects on neonates and postnatal animals, which seriously negatively affects livestock production. In this study, we aimed to identify potential genes associated with the diagnosis of IUGR through bioinformatics analysis. Based on the 73 differentially expressed related genes obtained by differential analysis and weighted gene co-expression network analysis, we used three machine learning algorithms to identify 4 IUGR-related hub genes (IUGR-HGs), namely, *ADAM9*, *CRYL1*, *NDP52*, and *SERPINA7*, whose ROC curves showed that they are a good diagnostic target for IUGR. Next, we identified two molecular subtypes of IUGR through consensus clustering analysis and constructed a gene scoring system based on the IUGR-HGs. The results showed that the IUGR score was positively correlated with the risk of IUGR. The AUC value of IUGR scoring accuracy was 0.970. Finally, we constructed a new artificial neural network model based on the four IUGR-HGs to diagnose sheep IUGR, and its accuracy reached 0.956. In conclusion, the IUGR-HGs we identified provide new potential molecular markers and models for the diagnosis of IUGR in sheep; they can better diagnose whether sheep have IUGR. The present findings provide new perspectives on the diagnosis of IUGR.

## 1. Introduction

Intrauterine growth restriction (IUGR) is a pregnancy-specific disease that occurs as a common perinatal complication in animal reproduction. IUGR can arise from mutations in survival conditions which result in fetal malformations and a mortality rate 12 times higher than that in normal neonates [1]. IUGR is characterized by the failure of the fetus and its organs to reach a normal physiological state in the uterus during pregnancy [2]. Fetal growth and development can be negatively influenced by many factors related to parental genetics and the living environment, which predominantly include nutrition management problems such as insufficient or excess nutrient supply or the malabsorption of nutrients [3]; environmental problems such as stress caused by maternal overcooling, overheating, or sudden weather changes [4]; and maternal developmental conditions such as abnormal development of the uterus and placenta and intrauterine disorders [4]. IUGR may be caused by one or more of these factors [5]. IUGR has a long-term negative impact on newborn and postnatal animals, often causing an increase in neonatal fetal mortality, slow growth and development, and increased disease rates in neonatal and later periods [6], which seriously affect animal husbandry production levels and lead to huge economic losses [7]. Therefore, a better understanding of the molecular mechanisms underlying IUGR is essential for effective diagnosis in sheep.

Abnormal placental implantation and dysfunction are the key mechanisms underlying IUGR [8]. Early placental development involves a combination of the proliferation, differentiation, and apoptosis of trophoblast cells [9]. The placenta is essential for normal fetal growth because it provides oxygen and nutrients to the developing embryo, producing the insulin-like growth factors involved in fetal growth [10]. Placental pathology affects both the mother and the fetus, and the pathogenesis of IUGR is associated with many molecular mechanisms. TGFβ, m-TOR proteins, NOD1 proteins, leptin, heat shock proteins, p53, glucocorticoids, and factors such as oxidative stress all play a significant role in IUGR and its associated diseases [11]. For example, the atypical expression of enzymes governed by TGFβ causes placental apoptosis, whereas the altered expression of TGFβ through hyper-alimentation causes impaired lung function [12]. Moreover, the cross-talk between cAMP and protein kinases plays a prominent role in the regulation of cortisol levels [13]. Furthermore, increased levels of NOD1 can lead to the development of IUGR by increasing the levels of inflammatory mediators [14] and increased leptin synthesis in the placental trophoblasts has been associated with IUGR [15].

IUGR may also change the liver weight, liver cell structure, and liver components of sheep, as well as hinder liver development and damage liver metabolism, immunity, and hematopoietic functions [16]. In late pregnancy, IUGR in sheep reduces the fetus’s weight, average daily gain, quality of the carcass and its components, and the density of muscle fibers [17]. Simultaneously, IUGR during late pregnancy affects the structure and thermogenic function of sheep fetal perirenal adipose tissue, resulting in a decrease in the number of fetal perirenal brown adipose tissue preadipocytes and blocking the cell cycle of fetal perirenal adipocytes to varying degrees [18]. In sheep, IUGR caused by placental insufficiency results in increased fetal ventricular protein-to-DNA ratios [19]. Recent attempts at the genetic selection and breeding of sheep have resulted in new breeds with increased litter size (up to six fetuses per ewe) but greatly reduced birth weight and lamb survival rates [20]. However, no method currently exists to effectively diagnose IUGR in sheep with multiple fetuses.

Therefore, identifying the potential molecular markers of IUGR in sheep and elucidating the molecular mechanisms of IUGR are essential for the diagnosis and prevention of IUGR in sheep. In this study, we combine multiple machine-learning algorithms to identify the potential molecular markers of IUGR in sheep, namely, IUGR-related hub genes (IUGR-HGs). We then incorporate the identified IUGR-HGs into the study model of an artificial neural network (ANN) for the early diagnosis and assessment of IUGR and explore the role of related differentially expressed genes (DEGs) in the pathogenesis of IUGR in sheep. Finally, we construct an IUGR score based on IUGR-HGs to assess the risk of IUGR. The findings of this study provide a theoretical and practical basis for the prediction and diagnosis of IUGR in sheep, as well as new perspectives for breeding disease resistance in sheep.

## 2. Materials and Methods

### 2.1. Data Acquisition and Preprocessing

We downloaded two IUGR sheep datasets from the GEO database (https://www.ncbi.nlm.nih.gov/geo/ accessed on 1 February 2023): GSE114580 and GSE64578. The GSE114580 dataset contained 12 sheep tissue samples: 6 IUGR samples and 6 normal samples. The GSE64578 dataset contained 11 sheep tissue samples: 6 IUGR samples and 5 normal samples. Both datasets were converted from probes to their corresponding gene symbols with reference to GPL2112 platform annotation information. We used the R package “limma [21]” to correct the mRNA expression data in both datasets. We then merged the normalized mRNA expression data of the two datasets and eliminated batch effects using the R package “sva [22]”.

### 2.2. Differential Analysis of Gene Expression

We identified the DEGs between the two groups by comparing the expression profile data from the IUGR samples and normal samples with the R package “limma [21].” The threshold value was |log2FoldChange| > 0.5 and the adjusted *p*-value was <0.05, which yielded 102 DEGs.

### 2.3. Weighted Gene Co-Expression Network Analysis

We performed a weighted gene co-expression network analysis (WGCNA) of 1818 genes in sheep samples using the R package “WGCNA” [23]. To ensure that the constructed co-expression network was close to the scale-free distribution, we selected six as the soft power when performing WGCNA on the samples to obtain five modules; we then calculated their relationships with normal and IUGR samples. Subsequently, we selected genes from the blue module. Finally, 459 genes in the blue module were intersected with 102 DEGs to identify 73 IUGR-related DEGs.

### 2.4. Protein–Protein Interactions between Differentially Expressed IUGR-Associated Genes

The STRING database (https://string-db.org/ accessed on 15 February 2023) and Cytoscape v3.9.1 software were used to analyze the protein–protein interactions (PPI) between the IUGR-related DEGs [24].

### 2.5. Functional Enrichment Analysis

We performed a functional enrichment analysis to better define the biological processes and functions in which these 73 differentially expressed IUGR-associated genes are enriched and to better understand the mechanisms of IUGR occurrence. Specifically, we performed gene ontology (GO) and Kyoto Encyclopedia of Genes and Genomes (KEGG) analyses of the 73 IUGR-related DEGs using DAVID (https://david.ncifcrf.gov/ accessed on 25 February 2023) online software, where *p*-values less than 0.05 were considered significant.

### 2.6. Selection of Hub Genes

We used three machine learning algorithms to identify the hub genes: support vector machine recursive feature elimination (SVM-RFE), random forest (RF), and least absolute shrinkage and selection operator (LASSO). SVM-RFE is a sequential backward selection algorithm based on the maximum-margin principle. The optimal solution is filtered by sorting the scores of each feature using the model training samples [25]. RF integrates multiple trees through ensemble learning; its basic unit is a decision tree, which belongs to the category of ensemble learning in machine learning [26]. LASSO is a linear regression method that uses L1 regularization to make some of the learned feature weights zero, thereby achieving sparsity and feature selection [27]. First, we used the SVM-RFE algorithm from the “e1071” package [28], the RF algorithm from the “randomForest” package [29], and the LASSO algorithm of “glmnet” package [30] to analyze 73 IUGR-related DEGs and identify potential candidate genes. We then intersected the candidate genes screened by the above three algorithms using Venn diagrams and identified four intersected hub genes (IUGR-HGs).

### 2.7. Consensus Clustering Analysis

We used the “ConsensusClusterPlus” package [31] to determine the molecular subtype of IUGR based on the four IUGR-HGs identified above. We used the PAM algorithm with Euclidean distance and performed 1000 iterations on the samples. The k values were increased from two to nine to determine the best clusters.

### 2.8. Construction of the IUGR Scoring System

To quantify the IUGR-related gene expression patterns of individual sheep, we constructed a scoring system to evaluate the IUGR-HG expression patterns of individual sheep, that is, the IUGR-HG signature, which we termed the IUGR score. By drawing on the methods of previous studies [32,33,34], we performed a principal component analysis based on the expression levels of the four IUGR-HGs and used principal component 1 and principal component 2 as the characteristic scores. The formula for calculating the IUGR score is as follows:IUGRscore = ∑(PC1_i_ + PC2_i_)
where “i” represents the IUGR-HGs. We grouped samples with IUGRscore > 0 as the high IUGRscore group and samples with IUGRscore ≤ 0 as the low IUGRscore group [35]. Finally, the receiver operating characteristic (ROC) curves were used to assess the accuracy of the IUGR score for determining the occurrence of IUGR.

### 2.9. Construction and Validation of an Artificial Neural Network (ANN) Model

We constructed an ANN model based on the IUGR-HGs using the R package “neuralnet”, which comprised three parts [36]: (1)An input layer, which included the gene expression of the four IUGR-HGs;(2)The first hidden layer, which included the gene expressions and weights of the four IUGR-HGs, and the second hidden layer, which included the weights of all the neurons in hidden layer 1;(3)The output layer, which indicated whether the sample was “normal” or “IUGR”.

To speed up convergence and improve the accuracy of the neural network, we set the number of neurons in the first hidden layer to eight and the number of neurons in the second hidden layer to three, and used the ROC to evaluate the prediction performance of the ANN according to training and test sets

### 2.10. Evaluation of the Diagnostic Value of the Selected Hub Genes in IUGR

We further investigated whether the selected IUGR-HGs were valuable for the diagnosis of IUGR in sheep. The performance of these genes was evaluated in the combined dataset using ROC curves. Specifically, we obtained hub gene expression data and disease status grouping data from samples of sheep with IUGR, performed ROC analysis using the “roc” function of the “pROC” package [37], and obtained the final area under the curve (AUC) results using the “ci” function. Figure 1 shows the complete workflow of this study.

### 2.11. Statistical Analysis

R v4.2.0 software was used for all statistical analyses and visualization. Multiple groups of data were statistically analyzed using analysis of variance [38], with comparisons between the two groups made using the Wilcoxon rank-sum test. For all statistical analyses, *p* < 0.05 was considered statistically significant.

## 3. Results

### 3.1. Screening for DEGs by Comparing IUGR and Normal Samples

First, we corrected and merged the gene expression profile data in the GSE114580 and GSE64578 datasets (Figure 2A). Next, we performed differential gene expression analysis by comparing the 12 IUGRs with 11 normal tissues in the combined dataset and obtained 102 differentially expressed genes (Appendix A), of which 50 were upregulated and 52 were downregulated (Figure 2B). These 102 DEGs were significantly differentially expressed between the IUGR and normal samples (Figure 2C).

### 3.2. Identification of Modular Genes Associated with IUGR by WGCNA

To identify the modular genes associated with IUGR in sheep, we performed WGCNA analysis on 1818 genes from sheep samples. We selected six as the appropriate soft strength (Figure 3A) and then performed clustering (Figure 3B) using a sample dendrogram and IUGR trait heat map showing the overall situation of normal and IUGR groups in the sample (Figure 3C). The selection of blue modules (Figure 3E) was determined by modular gene clustering (Figure 3D). We identified 459 genes associated with IUGR through WGCNA (Appendix A).

### 3.3. GO and KEGG Pathway Analysis of 73 DEGs

We then intersected the 102 DEGs with the 459 genes obtained by WGCNA and obtained 73 DEGs related to IUGR after the intersection (Figure 4A). PPI showed strong interactions (Figure 4B). These 73 potentially IUGR-related DEGs were used for subsequent analyses.

GO and KEGG analyses were used to clarify which biological processes and functions were enriched in these 73 DEGs and to better understand the pathogenesis of IUGR. The DEGs were annotated according to molecular function, cellular components, and biological process (GO categories). The GO enrichment analysis results (Appendix A) showed that the 73 DEGs were mainly enriched in “protein-lysine 6-oxidase activity”, “metalloendopeptidase inhibitor activity”, “hormone activity, serine-type endopeptidase inhibitor activity”, “negative regulation of membrane protein ectodomain proteolysis”, “peptidyl-lysine oxidation, cellular response to lipopolysaccharide”, “negative regulation of myeloid cell differentiation”, and other pathways (Appendix A). Among them, “protein-lysine 6-oxidase activity”, “metalloendopeptidase inhibitor activity”, and other pathways aggravate the degree of IUGR by regulating protein synthesis. The results of the KEGG enrichment analysis (Appendix A) showed that the 73 DEGs were mainly enriched in the “HIF-1 signaling pathway”, “metabolic pathways”, “p53 signaling pathway”, “glucagon signaling pathway”, “oocyte meiosis”, “biosynthesis of amino acids”, “glycosphingolipid biosynthesis—lacto and neolacto series”, “progesterone-mediated oocyte maturation”, “citrate cycle (TCA cycle)”, “TNF signaling pathway”, “lipid and atherosclerosis”, “fat digestion and absorption”, “vascular smooth muscle contraction”, “ovarian steroidogenesis”, “cellular senescence”, and other aspects (Appendix A). These pathways are associated with IUGR by limiting fetal development by altering the absorption of nutrients such as oxygen, sugars, and lipids.

### 3.4. Identification of Hub Genes via Machine Learning

Next, to identify the key IUGR-related DEGs, we trained and analyzed the 73 genes using RF, SVM-RFE, and LASSO algorithms which have been widely employed to analyze biological data and accurately identify hub genes in gene expression profiles. First, we utilized the LASSO algorithm to identify the variation in the regression coefficients of the 73 DEGs, selected the optimal and minimal criteria of the penalization parameter (λ) using 10-fold cross-validation (Figure 5A), and screened 10 candidate genes. We also established the SVM-RFE model to screen the nine candidate genes with the highest accuracy rate and the lowest error rate (Figure 5B). Furthermore, we incorporated the 73 DEGs into the RF model and screened out 25 candidate genes with an importance score > 0 (Figure 5C). Finally, we intersected the candidate genes screened by the above three algorithms (Table 1). *CRYL1*, *SERPINA7*, *ADAM9*, and *NDP52* were identified by all three machine learning approaches (Figure 5D) and so were defined as IUGR-HGs in this study.

### 3.5. Identification of Molecular Subtypes Based on IUGR-HGs and Verification of Molecular Subtypes Using the IUGR Score

To assess the differences between the different subtypes of IUGR, we identified the molecular subtypes of IUGR based on the four IUGR-HGs. Specifically, we clustered IUGR samples according to the transcriptome patterns of the four key genes using the k-means method of unsupervised consensus clustering. K = 2 was then selected as the optimal cluster number after comprehensive consideration (Figure 6A,B). IUGR samples were then divided into two subtypes (Figure 6C), that is, C1 and C2 subgroups, which had clear boundaries, suggesting stable and reliable clustering for the IUGR samples. Principal component analysis was further applied to validate the assignments of the two subtypes; the results showed that the samples in one subgroup were closer to each other than those in the other subgroup, suggesting that the two-dimensional principal component analysis distribution and the two subtypes had similar consistency (Appendix A). 

To quantify the personalized IUGR-related gene expression pattern of each sample more accurately, we constructed a scoring system, the IUGR score, based on the aforementioned four IUGR-HGs (*CRYL1*, *SERPINA7*, *ADAM9*, and *NDP52*). We calculated the IUGR score for each sample retrieved from the combined dataset using principal component analysis to assess the risk of IUGR. IUGRs retrieved from the combined dataset were classified into low or high IUGR subgroups according to an IUGR score <0 or >0, respectively (Appendix A). The basic profiles of each sheep sample were visualized with IUGR in alluvial plots, which included high- and low-IUGR score groups, as well as C1 and C2 subgroups from the cluster analysis (Figure 6D). The results showed that samples with a low IUGR score belonged to the normal group, whereas samples with a high IUGR score belonged to the IUGR group. A comparison of the IUGR score between the two groups (Figure 6E) confirmed that the IUGR group had higher IUGR scores, suggesting that the IUGR score can indicate the risk of IUGR. We also showed that the IUGR score increased sequentially from the normal to C2 to C1 groups (Figure 6F), indicating that their IUGR risk also increased in this order and revealing the relationship between gene expression and disease risk in IUGR subtypes (Figure 6H). Moreover, when the expression levels of *CRYL1*, *ADAM9*, and *NDP52* were high but the expression level of *SERPINA7* was low, the sheep samples belonged to group C1, which had the highest risk of IUGR (Appendix A). Finally, according to the ROC curves used to assess the accuracy of the IUGR score, the AUC value was 0.970, indicating good predictive performance (Figure 6G).

### 3.6. Construction and Validation of Artificial Neural Network (ANN) Models

The IUGR-HGs identified by the three machine learning algorithms can be used to classify IUGR tissues into two subtypes. Moreover, the IUGR score based on the IUGR-HGs can classify IUGR and normal tissues into high- and low-risk groups and identify high- and low-risk groups of IUGR molecular subtypes. Therefore, to verify the diagnostic performance of the IUGR-HGs, we used single IUGR-HGs to make diagnostic predictions using the combined dataset. The ROC curves of the dataset showed that the IUGR-HGs are good diagnostic targets for IUGR, with an AUC value of 0.992 for *CRYL1*, 0.992 for *NDP52*, 0.947 for *ADAM9*, and 0.955 for *SERPINA7* (Appendix A).

Subsequently, we incorporated the four IUGR-HGs into an ANN designed to predict whether a sheep sample has IUGR (Figure 7A). We then compared the ANN model prediction results with the actual grouping information of the samples. The ANN prediction results and accuracies for the training and test sets are listed in Table 2; the prediction accuracies were 1.000 and 0.833, respectively. Combining the training set and test set, the ANN accuracy reached 0.956. Finally, we used the ROC curves to evaluate the predictive ability of the ANN model for the training and test sets, with AUC values of 1.000 for the training set (Figure 7B) and 0.875 for the test set (Figure 7C). The ANN weights are shown in Appendix A.

## 4. Discussion

IUGR is a general term for impaired fetal development in the uterus of mammals, where the growth of the fetus and its organs is stunted because of a failure to provide adequate nutrients during gestation [39]. IUGR has a significant impact on the survival rate of individual offspring, fetal and postnatal growth performance, and the health status of sheep, pigs, rats, and other animals, as well as a substantial impact on livestock production and human health [40]. The clinical manifestations of IUGR include low birth weight and organ shrinkage of varying degrees; long-term effects can lead to impaired fetal development, lower metabolic capacity, and delayed neurological development, which greatly increase the risk of metabolic syndrome [41]. Therefore, exploring the potential biomarkers of IUGR and establishing IUGR-related prediction models are essential for the diagnosis and prevention of IUGR, the conservation of livestock and poultry genetic resources, and effective animal breeding.

In this study, we screened 102 DEGs by differential gene expression analysis and 459 IUGR-related genes by WGCNA, then intersected these two groups of genes to obtain 73 DEGs associated with IUGR. A functional enrichment analysis helped to identify the 20 most significantly enriched pathways related to the 73 DEGs, including the “HIF-1 signaling pathway”, “p53 signaling pathway”, and ”oocyte meiosis”. These processes are associated with IUGR. HIF-1α is a transcription factor that responds to changes in oxygen tension and is active in hypoxic environments (oas04066). Thus, oxygen, hypoxia, and HIF play crucial roles in placental formation and trophoblastic under-invasion [42]. HIF-1α can also regulate proliferation, apoptosis, and tolerance to hypoxia, with previous studies reporting that HIF-1α is associated with the pathology of IUGR [43]. p53 is an important transcription factor that plays a crucial role in determining the cellular stress response as it regulates key processes such as the apoptotic cascade and angiogenesis [44]. Uterine insufficiency and IUGR alter p53 CpG methylation, affect the mRNA levels of key apoptosis-related proteins, increase renal apoptosis, and reduce the number of glomeruli in the IUGR kidney [45]. These changes represent the mechanisms that contribute to the fetal origin of the disease. If oocyte meiosis is abnormal and the fertilized egg carries genetic material with chromosomal abnormalities, it may develop into an embryo and eventually a fetus which may lead to IUGR [46]. Taken together, these results suggest that the 73 genes are closely associated with IUGR, which confirms the validity of subsequent IUGR-HGs acquisition.

Among these 73 DEGs, a comparison of the key diagnostic genes identified by LASSO, SVM-REF, and RF revealed four IUGR-HGs: *ADAM9*, *CRYL1*, *NDP52*, and *SERPINA7*. *ADAM9* encodes a protease, which belongs to the ADAM (a disintegrin and metalloproteinase) family of proteins. It plays an important role in the nervous system, cardiovascular system, muscular system, and other tissues. In the nervous system, *ADAM9* is involved in processes such as neuron migration, axon growth, and synapse formation [47]. In the cardiovascular system, *ADAM9* is involved in the proliferation and migration of vascular endothelial cells [48]. In the muscle system, *ADAM9* is involved in muscle cell proliferation and differentiation [49]. *CRYL1* (crystallin lambda 1) is a protein-encoding gene. During embryonic development, *CRYL1* is involved in the formation of the nervous system. Studies have found that *CRYL1* is highly expressed in neuroepithelial cells during embryonic life and plays a regulatory role in the development of the nervous system. The deletion or mutation of *CRYL1* may lead to abnormalities in nervous system development [50]. In organogenesis, *CRYL1* is involved in kidney development. Studies have found that *CRYL1* plays an important role in the formation and maintenance of renal tubules, *CRYL1* participates in the normal development of renal tubules by regulating cell proliferation, migration, differentiation, and other processes [51]. *NDP52* is also known as *CALCOCO2* (calcium binding and coiled-coil domain 2) and encodes a protein containing a coiled-coil structural domain. *NDP52* binds to proteins such as ULK1 and TBK1 and participates in the regulation of the autophagy process. Autophagy is an intracellular waste removal process that is essential for the normal development and functional maintenance of cells. *NDP52* binds to proteins such as NF-κB and IRF3 and is involved in the regulation of inflammation and immune responses [52]. In addition, studies have found that *NDP52* plays an important role in cell migration and proliferation, participating in the remodeling of the cytoskeleton and the regulation of intracellular signaling [53]. This shows that *NDP52* has a positive significance on the regulation and development of the animal immune system. *SERPINA7* (serpin family A member 7) is a protein-encoding gene. Studies have shown that loss of SERPINA7 can lead to embryonic death or abnormal development. In the early stages of embryonic development, SERPINA7 can regulate the proliferation and differentiation of embryonic stem cells and promote embryonic development. In later embryonic stages, SERPINA7 can promote organ formation and tissue development, such as the heart, liver, and kidneys [54]. In addition, SERPINA7 can promote the differentiation and maturation of osteoblasts and enhance their bone matrix deposition ability. SERPINA7 can also inhibit the apoptosis of osteoblasts and promote bone growth and repair [55]. In summary, the four identified IUGR-HGs are involved in important processes of animal embryogenesis and fetal organ development in the maternal uterus. This also reflects the importance of IUGR-HGs.

We then identified two IUGR disease subtypes based on the four IUGR-HGs, called C1 and C2, and constructed an IUGR score to assess the IUGR risk based on the IUGR-HGs. The results showed that the IUGR risk was higher in the C1 subgroup than in the C2 subgroup and that the IUGR risk was higher in the IUGR group (C1 and C2) than in the normal group. This shows that the IUGR score can accurately identify the high-risk and low-risk groups of IUGR molecular subtypes. The higher the IUGR score, the higher the risk of IUGR. The accuracy of the IUGR score was also evaluated using the ROC curve, which showed the good prediction performance of the IUGR-HGs. This implies that IUGR-HGs may play a regulatory role in the development of IUGR and highlights their potential as diagnostic targets for IUGR. We also determined the relationship between IUGR-HG expression and IUGR risk, showing that the risk of IUGR was higher when *CRYL1*, *ADAM9*, and *NDP52* expression was high but *SERPINA7* expression was low. This also proves the feasibility of identifying molecular subtypes based on the identified IUGR-HGs.

In recent years, research on the application of ANNs with regard to diseases has increased dramatically. This advanced technology has shown excellent performance in diagnosis, prediction, and treatment, especially for pregnancy-related diseases [56]. ANN-based models generally exhibit the best accuracy and AUC values, with some achieving 100% accuracy [57,58]. Therefore, we used the IUGR-HGs to construct an ANN model to predict the occurrence of IUGR in the samples. The prediction accuracies for the training and test sets were 1.000 and 0.830, respectively, whereas the predictive ability of the ANN model on the training and test sets was 1.000 and 0.875, respectively. Overall, the results indicate that the ANN model is convincing; the IUGR-HGs have the potential to be used as independent diagnostic predictors of IUGR in sheep and the ANN developed can be used as an independent diagnostic model of IUGR in this study. This also shows that artificial intelligence has great development potential in the diagnosis and prevention of animal diseases.

Certainly, there are some limitations in the present study. Firstly, we constructed a diagnostic prediction model based on only 23 samples from the GEO database. Thus, a larger cohort of sheep samples is needed to confirm. Secondly, the accuracy of the ANN model needs further investigation, and more basic and clinical studies should be conducted to find more straightforward and cost-effective methods for diagnostics. In the future, we will focus on analyzing the genetic mutation data of IUGR to explore the changes in the IUGR disease-causing genes on the molecular level. Finally, in order to further reveal the potential regulatory role of IUGR-HGs, functional experiments will be required in the future.

## 5. Conclusions

Four potential sheep IUGR-related diagnostic genes, *ADAM9*, *CRYL1*, *NDP52*, and *SERPINA7*, can constitute an “IUGR marker” that can help distinguish whether sheep suffer from IUGR or not. A new ANN model based on the IUGR-HGs enables the accurate diagnosis of IUGR in sheep. This study provides a new perspective for further explaining the mechanism of sheep IUGR.

## Figures and Tables

**Figure 1 animals-13-03305-f001:**
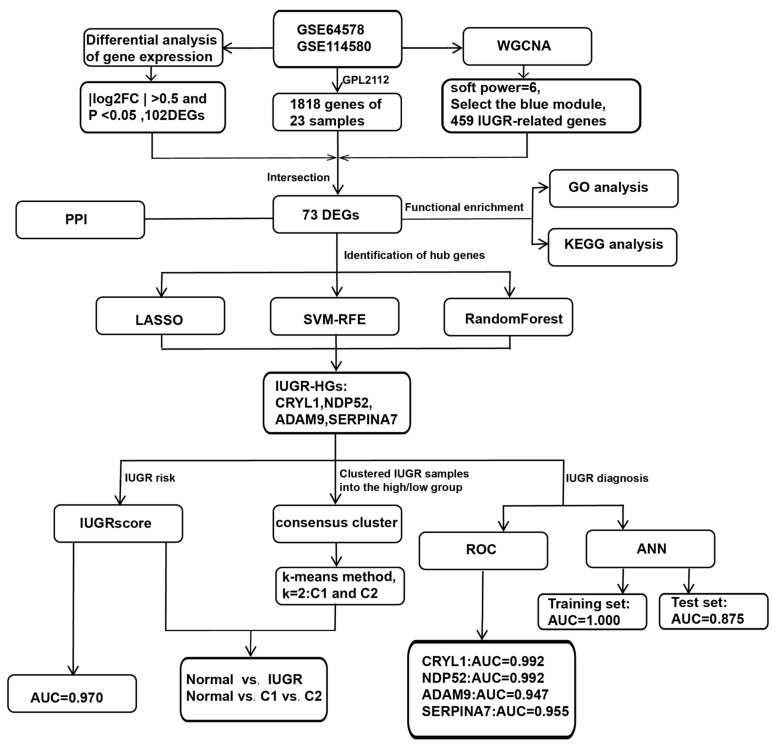
Workflow chart of this study.

**Figure 2 animals-13-03305-f002:**
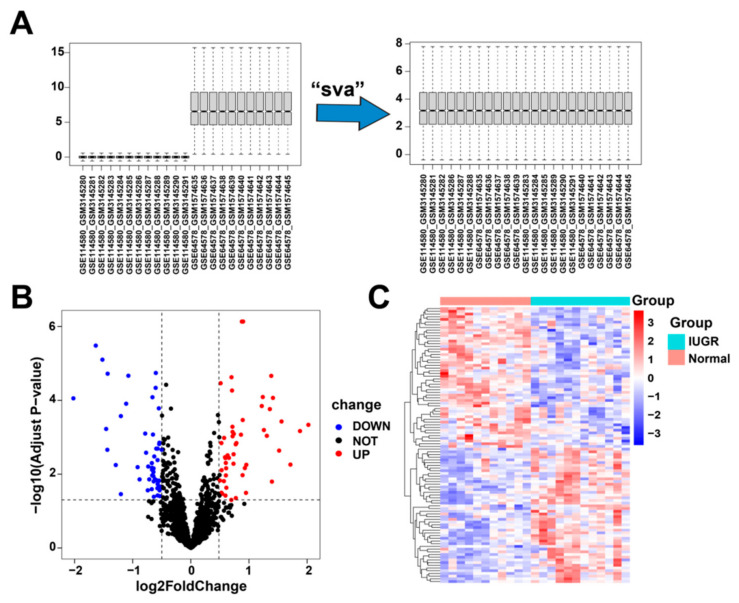
Data correction and screening of differentially expressed related genes in IUGR. (**A**) The overall landscape of data features before and after data merging. (**B**) Volcano plot showing the differentially expressed genes (DEGs) in the samples; red represents upregulation and blue represents downregulation. (**C**) Heatmap showing the overall landscape of the 102 DEGs between the normal and IUGR samples.

**Figure 3 animals-13-03305-f003:**
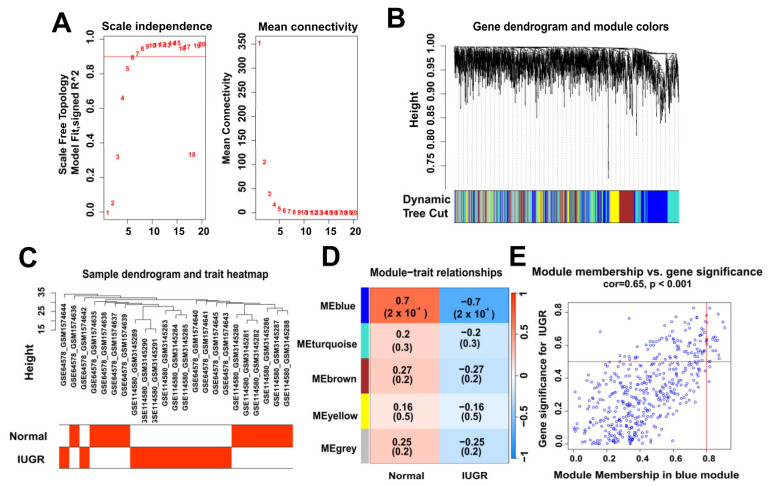
Identification of modular genes associated with IUGR by WGCNA. (**A**) Graphs of the soft-threshold power versus scale-free topology model fit index and mean connectivity. Six was chosen as the appropriate soft power. (**B**) Dendrogram of the genes clustered on the basis of a dissimilarity measure. (**C**) Dendrogram of samples and a heatmap plot of the indicated traits. (**D**,**E**) Analysis of relationships among gene modules and different traits; the blue module shows the most IUGR trait-specific modules in the samples.

**Figure 4 animals-13-03305-f004:**
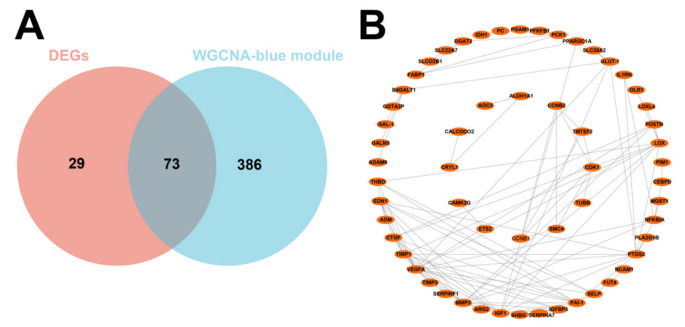
Screening of core differential genes in IUGR and its PPI network. (**A**) Venn plot showing the intersection of DEGs with blue module genes. (**B**) PPI network showing the interaction between 73 IUGR-related DEGs.

**Figure 5 animals-13-03305-f005:**
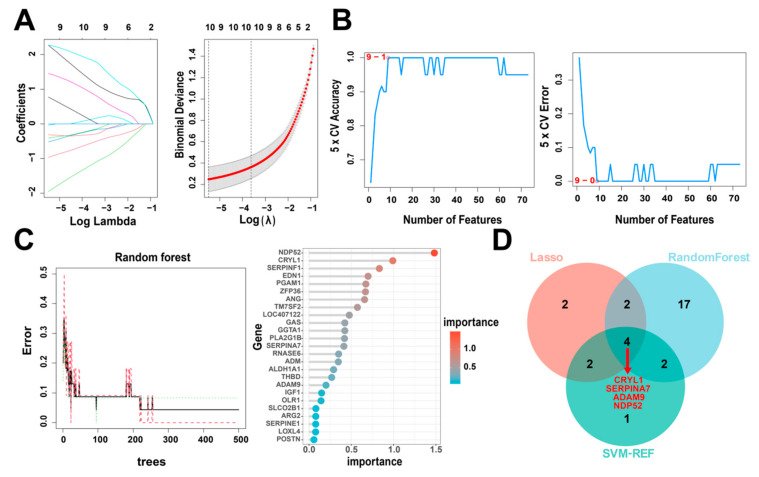
Identification of IUGR-HGs in IUGR. (**A**) LASSO coefficient profiles of candidate genes and the cross-validation for tuning predictor selection. (**B**) Nine potential genes were identified by the SVM-RFE algorithm with an accuracy of 1. (**C**) Random forest (RF) error rate versus the number of classification trees, and the gene importance scores of the RF model. (**D**) Venn plot showing the hub genes screened by the RF, SVM-RFE, and LASSO algorithms.

**Figure 6 animals-13-03305-f006:**
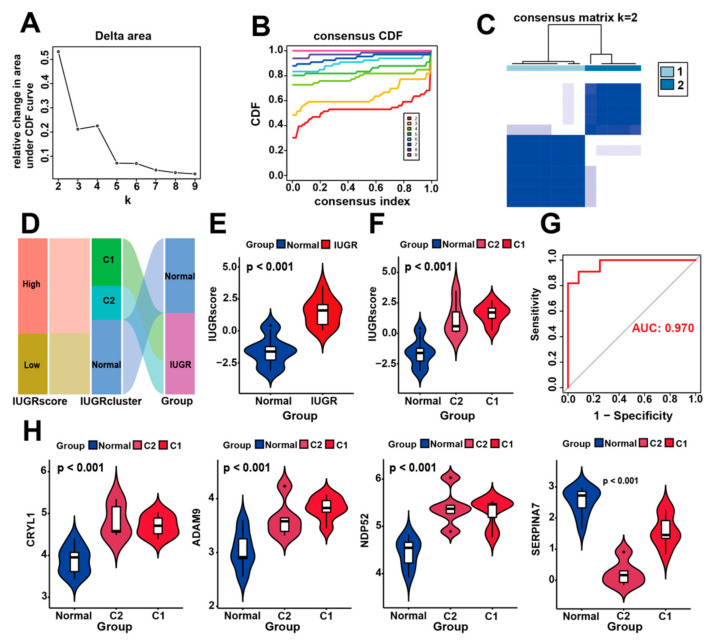
Construction of IUGR scoring system and identification of IUGR grouping subtypes. (**A**–**C**) Consensus clustering of sheep samples with IUGR, k = 2. (**D**) Alluvial plot showing the IUGR score, IUGR cluster, and disease changes. (**E**) Violin plot showing the IUGR score between the normal and IUGR groups. (**F**) Violin plot showing the IUGR score among the normal, C1, and C2 groups. (**G**) ROC curve of the IUGR score. (**H**) Expression analysis of the four hub genes in the three subgroups.

**Figure 7 animals-13-03305-f007:**
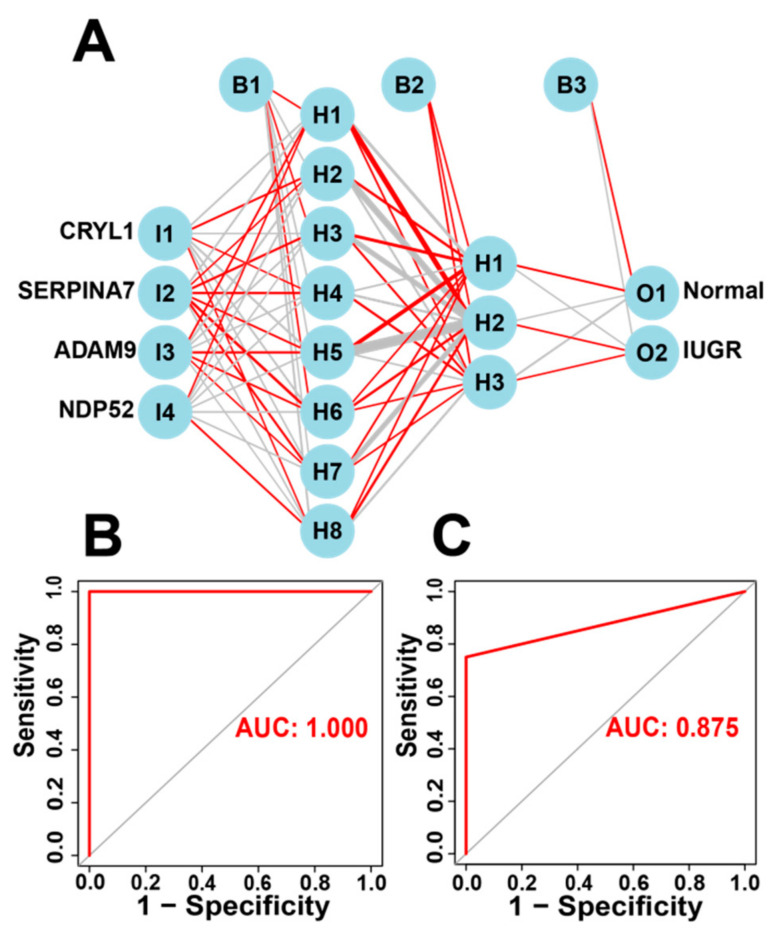
Construction of the ANN based on IUGR-HGs to diagnose normal and IUGR samples. (**A**) ANN construction process; (**B**) AUC of the training cohort with a value of 1; and (**C**) AUC of the test cohort with a value of 0.875.

**Table 1 animals-13-03305-t001:** The optimal candidate genes screened by three machine learning algorithms.

Methods	Genes
Lasso	*CRYL1*, *TM7SF2*, *SERPINA7*, *POSTN*, *ADAM9*, *CKM*, *LGALS1*, *CYB5*, *CCNB2*, and *NDP52*
RandomForest	*NDP52*, *CRYL1*, *SERPINF1*, *EDN1*, *PGAM1*, *ZFP36*, *ANG*, *TM7SF2*, *LOC407122*, *GGTA1*, *GAS*, *PLA2G1B*, *SERPINA7*, *RNASE6*, *ADM*, *ALDH1A1*, *THBD*, *ADAM9*, *IGF1*, *OLR1*, *SLCO2B1*, *ARG2*, *SERPINE1*, *LOXL4*, and *POSTN*
SVM-REF	*SERPINA7*, *CRYL1*, *NCAM1*, *NDP52*, *ZFP36*, *ADAM9*. *CYB5*, *GGTA1*, and *CKM*

**Table 2 animals-13-03305-t002:** Artificial neural network diagnosis effect for the training and test sets.

		Training Set		Test Set	
		Normal	IUGR	Normal	IUGR
Prediction	Normal	10	0	2	1
	IUGR	0	7	0	3
	Normal accuracy	1		1	
	IUGR accuracy	1		0.75	
	AUC	1		0.875	

## Data Availability

The datasets (GSE114580 and GSE64578) for this study can be found in the GEO Public Database (https://www.ncbi.nlm.nih.gov/geo/, accessed on 1 February 2023). All data generated or analyzed during this study are included in this article/additional files; further inquiries can be directed to the corresponding author.

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
