# Peer review of "Four Markers Useful for the Distinction of Intrauterine Growth Restriction in Sheep"

_animals, 2023, doi:10.3390/ani13213305_

Round 1

Reviewer 1 Report

In this manuscript, the authors found four potential sheep IUGR-related diagnostic genes and construct a new ANN model to distinguish and accurate sheep IUGR. on the IUGR-HGs enables accurately diagnose IUGR in sheep. This article has certain novelty to sheep IUGR diagnosis and provide a new perspective for further explaining the mechanism of sheep IUGR, so it could be considered to be published in this journal. However, there are still some specific comments before acceptance.

1.       Figure 4A is inconsistent, please shrink it.

2.       In 3.3 section, the authors can briefly describe the GO and KEGG pathway analysis of 73 DEGs, especially some important processes or pathways related with IUGR.

3.       I don’t understand why the author defined the grouped samples with IUGRscore > 0 as the high IUGRscore group and samples with IUGRscore ≤ 0 as the low IUGRscore group. If the IUGRscore is equal 0 or very close to 0, the difference is not significant.

Author Response

Point 1: Figure 4A is inconsistent, please shrink it.

Response 1: Dear reviewer, thanks for your correction. I have shrinked Figure 4A.

Point 2: In 3.3 section, the authors can briefly describe the GO and KEGG pathway analysis of 73 DEGs, especially some important processes or pathways related with IUGR.

Response 2: Dear reviewer, thank you for your comments. I have listed the pathways and briefly described them in 3.3 section. The description part is as follows:

  1. Among them, “protein-lysine 6-oxidase activity”, “metalloendopeptidase inhibitor activity”and other pathways aggravate the degree of IUGR by regulating protein synthesis.
  2. These pathways are associated with IUGR by limiting fetal development by altering the absorption of nutrients such as oxygen, sugars, and lipids.

I put this in the discussion in more detail.

Point 3: I don’t understand why the author defined the grouped samples with IUGRscore > 0 as the high IUGRscore group and samples with IUGRscore ≤ 0 as the low IUGRscore group. If the IUGRscore is equal 0 or very close to 0, the difference is not significant.

Response 3: Dear reviewer, thank you for your valuable suggestions.Using the median as a threshold is a very common analysis method. Many studies have adopted this method. The median of the IUGRscore is very close to 0, and in our cited literature, 0 is also used as the threshold. In our results, it is shown that the two groups are significantly different (including the part close to 0). These contents confirm that this analysis method has certain statistical significance. The reliability of risk scores also needs to be continuously verified and practiced in clinical practice.

Reviewer 2 Report

The conclusion is too short and incomplete. The authors may consider rewrite the manuscript for clearer and better readability. 

The quality of English language can be improved. For example, in line 26, the author stated that "affects the level of livestock production." which can be replaced as negatively affects livestock production. 

Author Response

Point 1: The conclusion is too short and incomplete. The authors may consider rewrite the manuscript for clearer and better readability.

Response 1: Dear reviewer, thank you for your correction. I will follow your suggestions and revise the content of this article. The amendments are as follows:

  1. 1.I have modified the abstract, reduced the length, and added a description of the results.
  2. 2.In the Results section 3.3, I added a brief description of some important processes or pathways related to IUGR
  3. 3.In the discussion section, I reorganized the logic of the entire discussion section.

For more details see the manuscript.

Point 2: The quality of English language can be improved. For example, in line 26, the author stated that "affects the level of livestock production." which can be replaced as negatively affects livestock production. 

Response 2: Dear reviewer, thank you for your correction. I have replaced "affects the level of livestock production." with "negatively affects livestock production". And we used Editage (www.editage.cn) services to help polish the article.

Reviewer 3 Report

The present research studies "to identify potential genes associated with the diagnosis of IUGR through bioinformatics analysis. The ANN model based on the IUGR-HGs enables accurate diagnosis of IUGR in sheep". This research provides interesting information. However, some important changes need to be made before final publication.

Abstract: review the "Journal" guidelines. It is mentioned in "MDPI Style Guide" the following: "The abstract contains a summary of the entire paper and can be up to 200 words long with only one paragraph". (https://www.mdpi.com/authors/layout) In this case it exceeds the number of words. I also recommend mentioning more about the results of this research. Therefore, restructure this section.

Line 26-27. “Currently, there are few diagnostic methods and treatment 26 options for IUGR, and its molecular mechanisms are not fully understood” this was already mentioned in line 15-16. I suggest eliminating it.

INTRODUCTION

I recommend mentioning "artificial neural network" (ANN) can be used as study models..

MATERIAL AND METHODS

A general comment regarding the design of this research, would this study be representative due to the number of samples obtained and, since there are few samples, could this not affect the genetic variability of any individual?

It would also be convenient to mention the characteristics of the sampled individuals, if they were from the same geographical region, ages, races, time of year, etc.

Line 156.- in section “Statistical analysis” mentioned “We used the PAM algorithm with Euclidean distance and performed 1,000 iterations on the samples”. why didn't they use 10,000 interations in the samples?

RESULTS

Line 323-324. “This implies that IUGR-HGs may 323 play a regulatory role in the development of IUGR and highlights their potential as diagnostic targets for IUGR.” I suggest moving to the "Discussion" section and explain its possible implications.

Line 377-338. “Overall, the results indicate that the ANN model is convincing and that IUGR-HGs have the potential to be used as independent diagnostic predictors of IUGR in sheep”. I suggest moving to the "Discussion" section and explain its possible implications.

DISCUSSION

I recommend restructuring this section according to the results presented and be more specific. That is, the effect of IUGRs as "disruption of the development of animal embryos or fetal organs in the maternal uterus" and these markers as a diagnostic tool for IUGRs.

Learn more about these genes, how they can affect embryo development and uterine implantation.

Also mention the implications of the "Construction of artificial neural network (ANN) models" as study models.

Line 407. I recommend that you learn more about the importance and relationship of the "73 IUGR-related SDRs".

Line 430. I recommend increasing the "IUGR risk" related to the "IUGR score" according to the identification of the two disease subtypes.

Line 408. “ADAM9, CRYL1, NDP52, and SERPINA7”. I recommend mentioning the pathways involved both metabolic, physiological, etc., that may be involved in these markers with IUGRs, on embryonic development and implantation.
